# Early-life disruption of amphibian microbiota decreases later-life resistance to parasites

Sarah A. Knutie [1,3], Christina L. Wilkinson[1], Kevin D. Kohl[2,4] & Jason R. Rohr[1]

Changes in the early-life microbiota of hosts might affect infectious disease risk throughout life, if such disruptions during formative times alter immune system development. Here, we test whether an early-life disruption of host-associated microbiota affects later-life resistance to infections by manipulating the microbiota of tadpoles and challenging them with parasitic gut worms as adults. We find that tadpole bacterial diversity is negatively correlated with parasite establishment in adult frogs: adult frogs that had reduced bacterial diversity as tadpoles have three times more worms than adults without their microbiota manipulated as tadpoles. In contrast, adult bacterial diversity during parasite exposure is not correlated with parasite establishment in adult frogs. Thus, in this experimental setup, an early-life disruption of the microbiota has lasting reductions on host resistance to infections, which is possibly mediated by its effects on immune system development. Our results support the idea that preventing early-life disruption of host-associated microbiota might confer protection against diseases later in life.

[1] Department of Integrative Biology, University of South Florida, Tampa, FL 33620, USA. [2] Department of Biological Sciences, Vanderbilt University, 465 21st Ave South, Nashville, TN 37235, USA [3] Present address: Department of Ecology and Evolutionary Biology, University of Connecticut, Storrs, CT 06269, USA [4] Present address: Department of Biological Sciences, University of Pittsburgh, Pittsburgh, PA 15260, USA. Correspondence and requests for materials should be addressed to  S. A.K. (email: saknutie@gmail.com)

A disruption of the normal microbiota of hosts just before parasite exposure has been shown to increase infection risk[1–5]. For example, at parasite exposure, an experimental reduction of bacterial diversity in mouse guts increased *Clostridium* infections relative to mice with normal gut microbiota[1]. One mechanism by which the bacterial community can protect their host from infections is through direct competition with the parasite[6, 7]. In contrast to this direct effect, host-associated microbiota may indirectly affect infections by influencing the maintenance or development of the immune system[8–10].

Early-life disruption of host-associated microbiota might increase, decrease, or have no effect on infection risk for hosts later in life. For example, reductions in *Bacteroides fragilis* in the gut of mice can adversely affect the development of the immune system[8, 9], suggesting increased infection risk. However, the consequences of these lasting changes on actual infections remain unclear; modifications to particular components of the immune system may not be relevant for fighting all types of pathogens. Conversely, studies show that an early-life disruption of host-associated microbiota can result in a hyper-reactive immune system that may increase the subsequent risk of immune-related, non-infectious diseases, such as allergies and autoimmune diseases (e.g., inflammatory bowel diseases)[11, 12]. In these cases, a hyper-reactive immune system attacks either innocuous antigens (i.e., allergens) or the host itself[13, 14]. Thus, an early-life disruption of the microbiota could also cause hyperimmunity to real infectious agents, reducing both infection and disease risk.

In our study, we show that an early-life disruption of the gut and skin bacterial communities of tadpoles affects later-life resistance to parasitic gut worms in adult frogs. Specifically, we show that tadpole bacterial diversity is negatively correlated with parasite establishment in adult frogs. Adult frogs that have reduced bacterial diversity as tadpoles have three times more parasites than adults that did not have their microbiota manipulated as tadpoles. In contrast, adult gut bacterial diversity during parasite exposure is not correlated with parasite establishment in adult frogs. Our results suggest that preventing early-life disruptions of host-associated bacterial communities might reduce infection risk later in life.

## Results

**Experimental design and background**. In this study, we investigated whether a disruption of the microbiota of hosts early in life increases or decreases host resistance to infections later in life. Here we define host resistance as defenses that reduce parasite fitness. Specifically, we experimentally manipulated the microbiota of Cuban tree frog tadpoles (*Osteopilus septentrionalis*) and then exposed them to parasites later in life. To manipulate the tadpole microbiota, we reared tadpoles in either natural pond water (control), autoclaved (i.e., sterile) pond water, sterile pond water and short-term antibiotics, or sterile pond water and long-term antibiotics ($n = 20$ tanks per treatment, four tadpoles per tank) (see Methods for details). The autoclaved pond water eliminated environmental sources of microbes, whereas the antibiotic cocktail (short-term: 24-h exposure, long-term: 4-week exposure) served to reduce the bacteria that had already colonized the tadpoles. All tadpoles were fed sterilized spirulina and fish flakes and all post-metamorphic frogs were housed individually in containers with sterile *Sphagnum* sp. moss and fed non-sterile crickets.

We used 16S ribosomal RNA (rRNA) gene sequencing to characterize bacterial diversity and community composition of both tadpoles (skin and gut) and adult frogs (gut; Fig. 1). We present Faith's phylogenetic diversity as our measure of alpha diversity in the main text because it accounts for phylogenetic differences among taxa. Results based on other alpha diversity measurements were consistent with those based on phylogenetic diversity (See Supplementary Table 1). For ease of interpretation, we refer to tadpoles as juveniles and post-metamorphic frogs as adults, but acknowledge that some of our post-metamorphic frogs might have still been developing.

We also tested whether early- or later-life host microbiota best predicted the ability of adult frogs to resist environmentally common, skin-penetrating, gut worms *Aplectana hamatospicula* (Ascaridida: Cosmocercidae; Fig. 1). *A. hamatospicula* has a direct life cycle. Juvenile larvae penetrate the skin of frogs and then, in approximately 3 weeks, establish, mature and reproduce in the gastrointestinal (GI) tract[15, 16]. Worm eggs and larvae (they are ovoviviparous) are defecated by frogs, and after approximately a week of development, juvenile worms can infect the next host. We quantified host resistance of adult frogs at both the skin penetrating and gut establishment stages of the parasite, which occurred 5 months after the microbiota of juveniles were characterized (6 months after water treatments started). To determine the relationships among water treatment, tadpole and adult bacterial diversity, and adult resistance to parasitism, we employed structural equation modeling (SEM). We used a hypothetico-deductive approach, comparing eight a priori hypothesized path models based on Akaike information criterion (AIC; Supplementary Fig. 1, Supplementary Table 2).

**Effect of water treatment on host health**. Water treatment significantly affected juvenile survival (Coxme, $\chi^2 = 16.69$, df = 3, $P < 0.001$; Supplementary Table 3 and time to metamorphosis (GLMM, $\chi^2 = 120.85$, df = 3, $P < 0.0001$). Juveniles from the long-term antibiotic treatment took twice as long to metamorphose and had lower survival compared to juveniles from the control treatment and other manipulated water treatments (sequential Bonferroni post hoc multiple comparison test: $P < 0.05$ for long-term antibiotic frogs compared to all other treatments). However, mass at metamorphosis did not differ significantly among treatments (GLMM, $\chi^2 = 5.03$, df = 3, $P = 0.17$). In contrast, juvenile water treatment affected adult mass when measured 2 months after metamorphosis (GLMM, $\chi^2 = 19.44$, df = 3, $P < 0.001$); adults exposed to long-term antibiotics weighed less than frogs from the control treatment and other manipulated water treatments ($P < 0.05$ for long-term antibiotic frogs compared to all other treatments). Water treatment did not have a lasting effect on adult survival (Coxme, $\chi^2 = 5.68$, df = 3, $P = 0.13$).

**Effect of water treatment on microbiota**. Juveniles exposed to sterile pond water or sterile water plus antibiotic treatments had reduced gut and skin bacterial diversity (Fig. 2a and b; see Supplementary Table 1 for other alpha diversity metrics) and altered bacterial community membership (Fig. 2d and e) and structure (gut: permutational multivariate analysis of variance (PERMANOVA), $F_{3,35} = 3.36$, $P = 0.001$; skin: $F_{3,37} = 2.10$, $P = 0.02$) relative to juveniles exposed to non-sterile pond water. Specifically, juvenile water treatment significantly reduced the relative abundance of several phyla, such as Fusobacteria in the guts ($\chi^2 = 18.82$, df = 3, $P < 0.001$; Fig. 2d) and on the skin of juveniles ($\chi^2 = 12.32$, df = 3, $P = 0.006$; see Fig. 3 for all phyla). In contrast, adult gut bacterial diversity (Fig. 2c, Supplementary Table 1) and community membership (Fig. 2f) and structure (PERMANOVA $F_{3,76} = 1.42$, $P = 0.03$) were similar across treatments with the exception of adults exposed to sterile water with long-term antibiotics as juveniles being different from the other three treatments (Bonferroni post hoc multiple comparison test, $P < 0.01$).

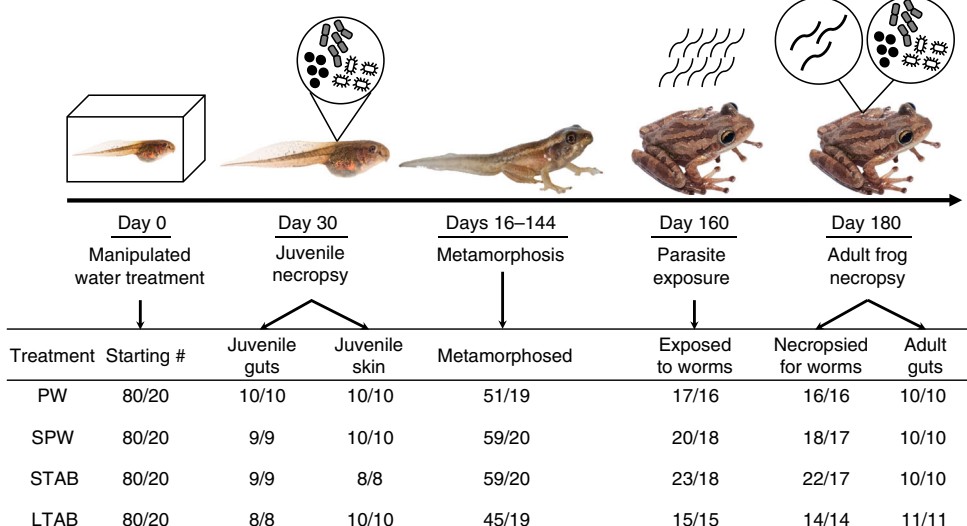

| Treatment | Starting # | Juvenile guts | Juvenile skin | Metamorphosed | Exposed to worms | Necropsied for worms | Adult guts |
|---|---|---|---|---|---|---|---|
| PW | 80/20 | 10/10 | 10/10 | 51/19 | 17/16 | 16/16 | 10/10 |
| SPW | 80/20 | 9/9 | 10/10 | 59/20 | 20/18 | 18/17 | 10/10 |
| STAB | 80/20 | 9/9 | 8/8 | 59/20 | 23/18 | 22/17 | 10/10 |
| LTAB | 80/20 | 8/8 | 10/10 | 45/19 | 15/15 | 14/14 | 11/11 |

**Fig. 1** Testing the effects of early-life disruption of the microbiota on later-life resistance to infections. Juveniles were reared in one of four water treatments: pond water control (PW), sterile pond water only (SPW), sterile pond water and short-term antibiotics (STAB), or sterile pond water and long-term antibiotics (LTAB). Adult frogs were exposed to *A. hamatospicula* worms to quantify the lasting effect of the manipulated microbiota of juveniles on adult resistance to infections. Numbers represent sample sizes (i.e., number of individuals/number of tanks) for each time point. Photos by Mark Yokoyama

**Effect of water treatment on host resistance.** Adult resistance to worm penetration was not affected significantly by any of the juvenile water treatments (Fig. 4a); however, frogs exposed to any of the three manipulated water treatments were three times more susceptible to worm establishment in their guts, and thus less resistant to infection, compared to frogs reared in natural pond water (Fig. 4b). In addition, adult resistance to infection could not be explained by juvenile mass at metamorphosis (GLMM, $\chi^2 = 0.00$, df = 1, $P = 0.98$), days to metamorphosis ($\chi^2 = 0.19$, df = 1, $P = 0.66$), or adult mass ($\chi^2 = 0.92$, df = 1, $P = 0.34$). Parasite treatment did not significantly affect bacterial diversity of adults (phylogenetic diversity: $\chi^2 = 0.10$, df = 1, $P = 0.75$, Shannon index: $\chi^2 = 0.11$, df = 1, $P = 0.74$, observed operational taxonomic units (OTUs): $\chi^2 = 1.81$, df = 1, $P = 0.18$, equitability: $\chi^2 = 0.00$, df = 1, $P = 0.98$).

**Structural equation model.** Given the convergence of the gut microbiota in adults across water treatments, the effect of juvenile water treatments on adult host resistance must be mediated by the microbiota of juveniles (Supplementary Fig. 2). This conclusion is supported by our SEM, which shows that the early-life reduction of host bacterial diversity associated with the water treatments ($P < 0.0001$) seemed to drive the increase in worm establishment later in life ($P = 0.002$; Fig. 5; Supplementary Fig. 1; Supplementary Table 2). In contrast, adult bacterial diversity during parasite exposure did not affect worm establishment in the SEM. Because the long-term antibiotic treatment affected the growth and survival of frogs (Supplementary Table 3), we included two additional SEM analyses: (1) without samples from the long-term antibiotic treatment (Supplementary Table 4), and (2) with samples from the long-term antibiotic treatment but also additional models with the effect of adult mass and tank-level survival on adult resistance to parasites (Supplementary Table 5). In these additional analyses, the top model remains the same as in the original SEM (Supplementary Table 2).

**Bacterial taxa and host resistance.** Relative abundance of phylum Fusobacteria was negatively related to the number of worms in the guts of adults (guts: $\chi^2 = 5.61$, df = 1, $P = 0.01$; skin: $\chi^2 = $ 2.65, df = 1, $P = 0.10$). Specifically, as the relative abundance of genus *Cetobacterium* increased in juveniles, infection risk decreased in adults (guts: $\chi^2 = 4.99$, df = 1, $P = 0.03$; skin: $\chi^2 = 7.44$, $P = 0.006$). See Supplementary Table 6 for the relationships between relative abundances of genera in juveniles and adults and parasite establishment in adults.

**Discussion**

Our results, using an amphibian model, show that an early-life disruption in host-associated microbiota can increase infection risk later in life. Specifically, bacterial diversity and relative abundance of phylum Fusobacteria (including genus *Cetobacterium*) in juveniles negatively correlated with infection risk in adult frogs. Moreover, bacterial diversity in adult frogs at the time of parasite exposure was not correlated with host resistance, in contrast with results of previous studies in mice and insects[1–3]. Our results suggest that an early-life disruption of the host-associated microbiota may affect the development of host resistance mechanisms, such as immunity, and has enduring effects on infection susceptibility later in life.

We think that the microbiota of juveniles likely played a role in priming the immune system against parasite establishment. We found that the relative abundance of certain bacteria phyla, such as Fusobacteria, in juveniles was positively correlated with parasite resistance in adulthood. We speculate that treatment-induced reductions of Fusobacteria may have disrupted immune system development, leading to decreased resistance to infection later in life. Interestingly, studies show that germ-free mice devoid of Fusobacteria exhibit lower IgG antibody production to pathogens when compared to conventional mice[17]. The analogous antibody in frogs (IgY) is produced in response to *A. hamatospicula* establishment in their guts[16], providing a candidate immune mechanism for our results that can be explored in future studies. Other potential resistance mechanisms could be identified by characterizing host gene expression (e.g., using RNA-seq) of candidate immune genes in both juveniles and adults[18].

The microbiota of adult frogs at the time of parasite exposure did not affect host resistance. In contrast, previous studies on mice and insects have found that bacterial communities help defend

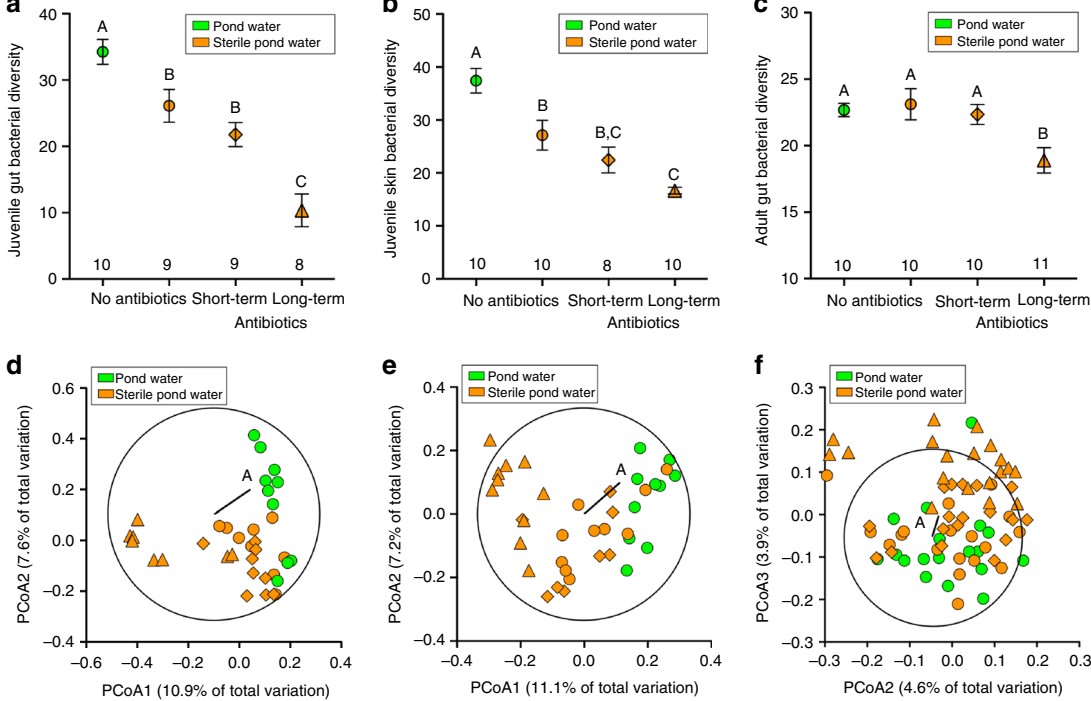

**Fig. 2** Effect of early-life microbiota disruption on juvenile and adult bacterial diversity and community membership. **a–c** Mean alpha Faith's bacterial diversity (phylogenetic diversity metric) across water treatments for samples from **a** juvenile guts (GLM, $\chi^2 = 62.99$, df = 3, $P < 0.0001$), **b** juvenile skin ($\chi^2 = 49.96$, df = 3, $P < 0.0001$), and **c** adult guts (GLMM, $\chi^2 = 13.86$, df = 3, $P = 0.003$). Error bars indicate the s.e.m. Numbers above the tick marks on the x-axis are the number of replicates (tanks) per treatment, and treatments that do not share letters above the error bars are significantly different based on a sequential Bonferroni post hoc multiple comparison test ($P < 0.05$). **d–f** Effect of water treatment on bacterial community membership from **d** juvenile guts (PERMANOVA, $F_{3,35} = 2.43$, $P = 0.001$), (**e**) juvenile skin ($F_{3,37} = 2.35$, $P = 0.001$), and **f** adult guts ($F_{3,76} = 1.42$, $P = 0.001$). Principal coordinates analyses (PCoA) were based on unweighted UniFrac scores. Vector A represents the direction and strength of the correlation between water treatment and relative abundance of phylum Fusobacteria and the *circles* represent a unit circle (radius = 1) to indicate the direction and correlational strength of vector A. For all panels, *different shapes* (antibiotic exposure) and *colors* (water treatment exposure) represent the different water treatments: pond water (*green circles*), sterile pond water only (*orange circles*), sterile pond water and short-term antibiotic water (*orange diamonds*), and sterile pond water and long-term antibiotic water (*orange triangles*)

their hosts against parasites at time of exposure by creating unfavorable conditions, maintaining the immune system against intruders, or outcompeting them[1–5]; our study suggests that the adult frog microbiota did not affect any of these conditions for worm establishment. We found that variation in bacterial diversity across treatments decreased after metamorphosis (i.e., bacterial diversity became more homogeneous after metamorphosis) (Fig. 2c), which is likely the result of either frog metamorphosis[19] or the adult diet consisting of non-sterile crickets reducing the treatment-induced variation in the microbiota that existed in the juvenile environment. The relatively minor variation in the bacterial diversity of adults was probably not enough to differentially affect worm fitness. Alternatively, the large impact of an early-life disruption to the microbiota may have overshadowed any effect of adult microbiota on parasite susceptibility.

Absolute abundance of bacterial taxa might also play a role in infection risk, but this aspect was not investigated in our study. A common method to quantify absolute abundance is quantitative PCR to determine the copies of 16S rRNA per gram of gut contents. However, we extracted bacterial DNA from the entire gut (intestines and contents) to maximize biomass recovery and therefore we did not specifically weigh the contents of the gut. Future studies could include absolute abundance data by removing, weighing, and extracting bacterial DNA from only the contents of the intestines. In addition, our study focused on bacterial communities, but we acknowledge that other microbial groups, such as fungi, archaea, protists, and viruses, were likely

altered by our experimental treatments and may have affected infection risk, which should also be explored in the future.

Our work supports the idea that there are crucial windows in development during which microbiota disruption may be particularly costly, having adverse persistent effects on infection risk. Although our study focused on frogs, we speculate that early-life microbiota disruption in humans might result in some similar effects, because of the known similarities in general microbiota composition and immune systems of frogs and mammals[20]. That is, an early-life disruption of human microbiota might stimulate an under-reactive immune response to infections, in addition to the previously established over-reactive immune response to innocuous agents (allergens and host)[13]. Furthermore, several factors, such as pollutants (including antibiotics)[2, 21, 22], nutrition[23–25], and climate[26] can disrupt the microbiota of animal hosts, and as a consequence, they might affect infectious disease risk[27, 28].

## Methods

**Experimental protocol.** Eighty 4L tanks received four *O. septentrionalis* tadpoles (juveniles) each, which were collected from pools at the University of South Florida (USF) Botanical Gardens, Hillsborough County, FL, USA (scientific collecting permit #LSSC-15-00014). All animal care procedures were in accordance with University of South Florida's Institutional Animal Care and Use Committee (IACUC protocol #IS00001610). The mean (±s.e.m.) Gosner stage for juveniles at the beginning of the experiment was 32.10 (±1.10); Gosner stage represents the development stage of advancement toward metamorphosis in tadpoles[29]. Juveniles were assigned randomly to one of four water treatments (20 tanks per treatment):

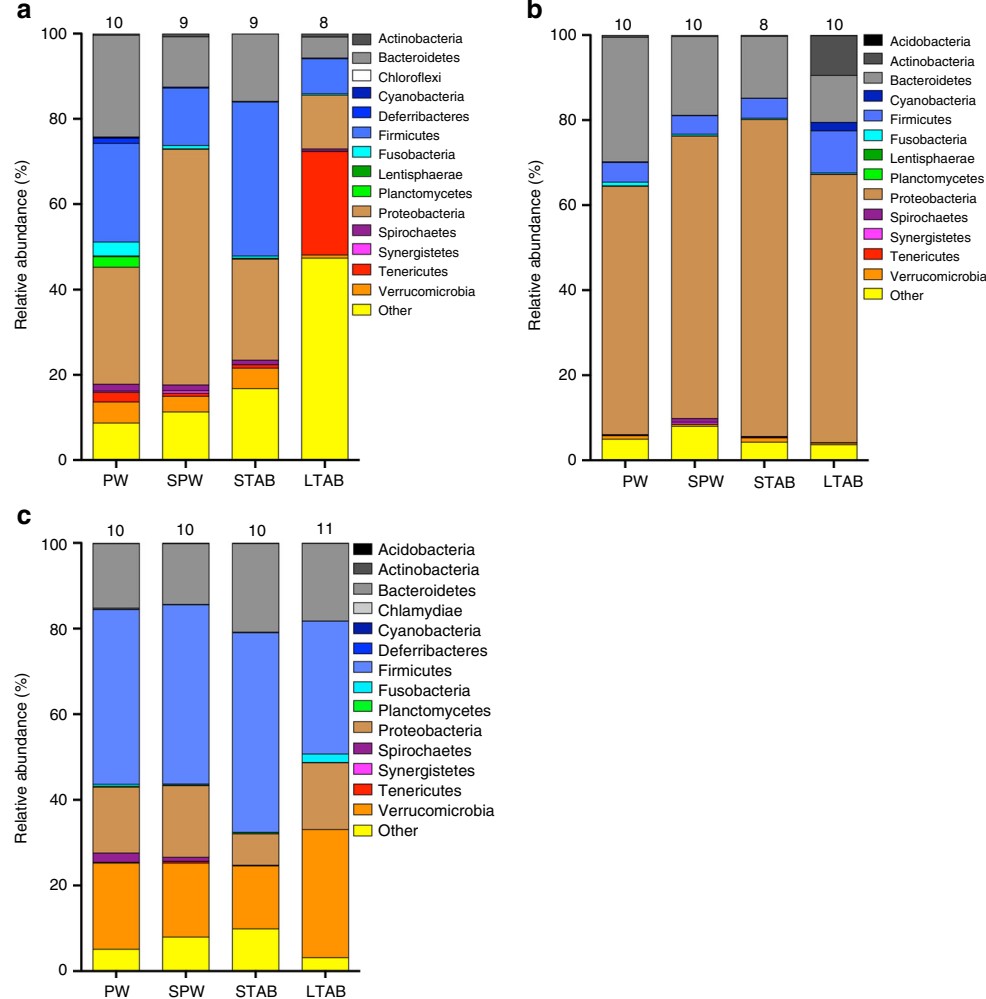

**Fig. 3** Effect of juvenile water treatment on relative abundance of bacterial phyla. We analyzed the bacterial community of the gut **a** and skin **b** of juveniles, as well as that of the gut of adults **c**. Water treatments include pond water (PW), sterile pond water only (SPW), sterile pond water and short-term antibiotics (STAB), or sterile pond water and long-term antibiotics (LTAB). For juvenile guts **a**, mean relative abundance of phylum Proteobacteria was three times higher for individuals from sterile pond water only (GLM, $\chi^2 = 25.09$, df = 3, $P < 0.0001$), Firmicutes was three times higher for individuals from pond water and short-term antibiotics ($\chi^2 = 13.92$, df = 3, $P = 0.003$), and Chloroflexi was higher for individuals from pond water ($\chi^2 = 25.11$, df = 3, $P < 0.0001$) compared to individuals from the other treatments. For juvenile skin **b**, relative abundance of phylum Actinobacteria was 20 times higher for individuals from long-term antibiotic water compared to the other three treatments ($\chi^2 = 70.42$, df = 3, $P < 0.0001$) and Bacteroidetes was three times higher for individuals from pond water compared to individuals from long-term antibiotic water ($\chi^2 = 11.89$, df = 3, $P = 0.008$). For adult guts, relative abundance of phylum Actinobacteria was five times higher for adults reared in sterile pond water compared to adults reared in long-term antibiotic water (GLMM, $\chi^2 = 10.05$, df = 3, $P = 0.02$), Spirochaetes was generally higher in adults reared in pond water but was only significantly higher compared to adults reared in short- and long-term antibiotic water ($\chi^2 = 14.39$, df = 3, $P = 0.002$), and Cyanobacteria was lower for adults reared in long-term antibiotic water compared to adults reared in pond water and short-term antibiotic water ($\chi^2 = 12.23$, df = 3, $P = 0.007$). Numbers above the bars are the number of replicates (tanks) per treatment

pond water (PW), sterile (autoclaved) pond water only (SPW), sterile pond water and short-term antibiotics (STAB), or sterile pond water and long-term antibiotics (LTAB). Juveniles from the short-term and long-term antibiotic treatments were exposed to a cocktail of antibiotics (30 mg/L enrofloxacin (Fluka, Sigma, St. Louis, MO, USA), 13.3 mg/L sulfamethazine and 2.67 mg/L trimethoprim (Thomas Labs, Fish Sulfa Forte, Tolleson, AZ, USA), and 5000 μg/L streptomycin and 5000 I.U./L penicillin (Mediatech, Inc., Manassas, VA, USA)) in sterile pond water for 24 h and 4 weeks, respectively; these were half the recommended doses from Holden et al.[30]. For the long-term antibiotic treatment, the antibiotic cocktail was added to the tank during each weekly water change. For sterilization, pond water was autoclaved in 10L carboys for 60 min at 121 °C.

Tanks were maintained in the laboratory (12 h light cycle, air temperature: 22 °C, water temperature: 22.6 °C, pH: 7.7, dissolved oxygen: 5.9 mg/L, nitrates: 0.96 mg/L) and all juveniles were fed ad libitum with a mixture of sterilized spirulina (NOW foods, Bloomingdale, IL, USA) and fish flakes (Omega One, Sitka, AK, USA) in an agarose block (autoclaved for 15 min at 121 °C). Juvenile survival was checked daily and water was changed weekly. After 4 weeks in the water treatments, 10 juveniles from each water treatment were euthanized to characterize

their bacterial community (Fig. 1). Their skin was swabbed with sterile cotton swabs to characterize skin bacteria and then juvenile GI tracts were removed to characterize their gut bacteria. Skin swabs and guts were frozen at −80 °C until DNA extractions.

The remaining juveniles in each tank were allowed to metamorphose. Individuals with all four limbs were removed from the tanks daily, weighed, and placed in cups (6 cm high × 12 cm diameter) with sterile organic *Sphagnum* sp. moss (Mosser Lee, Millston, WI, USA) (autoclaved for 30 min at 121 °C). The adults were maintained in the laboratory (12 h light cycle, 22 °C) on vitamin- and mineral-dusted crickets (vitamins and minerals: Rep-Cal Herptivite Multivitamin powder and Rep-Cal Calcium with Vitamin D3 powder, Los Gatos, CA, USA; crickets: *Acheta domesticus*, Armstrong's Cricket Farm, Glenville, GA, USA) (fed ad libitum) and survival was checked daily.

Approximately 2 months after metamorphosis (mean ± s.e.m. = 63.21 ± 2.75 days), frogs were weighed and then exposed to *A. hamatospicula* worms, which were collected from naturally parasitized adults at Flatwoods Park, Hillsborough County, FL, USA. Frogs were placed individually in parafilm-sealed petri dishes (100 mm diameter) with an air hole at the top of the lid; frogs were

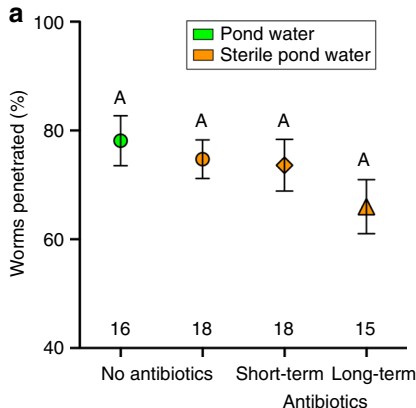
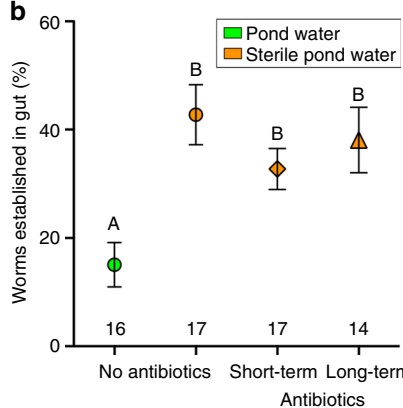

**Fig. 4** Effect of juvenile water treatment on parasite penetration and establishment in adults. **a** Mean percent (%) of *A. hamatospicula* worms that penetrated the skin of adult frogs (GLMM, $\chi^2 = 3.08$, df = 3, $P = 0.38$). **b** Mean percent (%) of *A. hamatospicula* worms that established in the guts of adult frogs ($\chi^2 = 29.11$, df = 3, $P < 0.0001$). Error bars indicate the s.e.m. Numbers directly above the x-axis tick marks are the number of replicates (tanks) per treatment. Treatments that do not share letters above the error bars are significantly different based on a sequential Bonferroni post hoc multiple comparison test ($P < 0.05$)

then exposed to worms (see Fig. 1 for sample sizes) or sham-exposed (PW: $n = 17$ individuals from 13 tanks; SPW: $n = 20/18$; STAB: $n = 22/17$; LTAB: $n = 16/14$) by pipetting 10 infectious larval *A. hamatospicula* worms in 3 mL of autoclaved pond water or 3 mL of autoclaved pond water without worms through the hole in the lid, respectively. After 24 h in the petri dish, we returned frogs to their individual cups with sterile *Sphagnum* sp. moss and used a dissecting microscope to count the worms remaining in the petri dish to determine the number of worms that penetrated each frog. Three weeks after exposure to *A. hamatospicula*, frogs were euthanized and necropsied to count the number of worms that established in their GI tracts. Frog guts were then collected from a subsample of individuals (sham-exposed: PW: $n = 10$ individuals, SPW: $n = 10$, STAB: $n = 10$, LTAB: $n = 9$; see Fig. 1 for sample sizes for the parasitized treatment) and frozen at −80 °C until DNA extractions.

We also determined whether juvenile worms could be observed on the surface of the frog's skin without penetrating the skin successfully after parasite exposure. We exposed five additional frogs to juvenile worms using the methods described above. After 24 h, we anesthetized frogs and scanned the surface of the skin for visible worms that did not penetrate it successfully. We did not observe any juvenile worms on the surface of the skin, which suggests that if worms were not found in the petri dish, they penetrated the host successfully.

**Bacterial DNA extraction and sequencing**. We isolated total DNA from frog guts and skin using a MoBio PowerFecal DNA Isolation Kit; DNA extracts were then sent to Argonne National Labs for sequencing. We also extracted and sequenced "blank" samples, which were collected using sham-necropsies and sham-extractions (i.e., without an experimental sample) to control for methodological contamination[31]. Bacterial inventories were conducted by amplifying the V4 region of the 16S rRNA gene using primers 515F and 806R and paired end sequencing on an Illumina MiSeq platform[32]. Sequences were analyzed using QIIME version 1.9.1[33]. We applied standard quality control settings and split sequences into libraries using default parameters in QIIME. Sequences were grouped into OTUs using pick_open_reference_otus.py with a minimum sequence identity of 97%. The most abundant sequences within each OTU were designated as a "representative sequence" and aligned against the Greengenes core set[34] using PyNAST[35] with default parameters set by QIIME. A PH Lane mask supplied by QIIME was used to remove hypervariable regions from aligned sequences. A phylogenetic tree of representative sequences was built using FastTree[36]. OTUs were classified taxonomically using UCLUST[37] with the reference Greengenes database[34]. Singleton OTUs and sequences identified as chloroplasts or mitochondria were removed from the analysis. In addition, any OTUs present in the "blank samples" were considered contaminants and were removed from all other samples[31]. Contaminant OTUs were largely similar to those presented in Salter et al.[31].

Several measurements of alpha diversity were calculated. We calculated the number of observed OTUs, equitability, the Shannon index, and Faith's phylogenetic diversity[38], the latter of which measures the cumulative branch lengths from randomly sampling 1900 sequences from each sample (the minimum number of sequences returned from each sample). For each sample, we calculated the mean of 20 iterations. We calculated unweighted and weighted UniFrac distances between samples in QIIME using 1900 sequences for bacterial community composition analyses.

**Statistical analyses**. We determined the effect of water treatment on juvenile bacterial diversity using generalized linear models (GLMs) with Gaussian errors. To determine the effect of juvenile water treatment on adult bacterial diversity, mass at metamorphosis, days to metamorphosis, and adult mass, as well as the

effect of parasitism on adult bacterial diversity, we used generalized linear mixed models (GLMMs) with Gaussian errors and tank as a random effect. Juvenile and adult samples were collected from different individuals within the same replicate (tank) because juvenile sampling required destructive sampling. Therefore, the juvenile and adult samples within tanks were paired in the analyses. Gaussian analyses were conducted using the glm (GLM) and lmer (GLMM) functions with the lme4 package. We determined the effect of water treatment on juvenile and adult survival using a censored Cox mixed effects model with the coxme function. Probability values were calculated using log-likelihood ratio tests using the Anova function in the car package. GLM, GLMM, and survival analyses were conducted in RStudio (2013, version 0.98.1062). All figures were made in Prism (2008, version 5b).

We determined the effect of juvenile water treatment on bacterial community membership (unweighted) and structure (weighted) using PERMANOVA + (with 999 permutations) in PRIMER (2008, version 6.1.11). Bonferroni post hoc multiple comparison tests were used to compare bacterial communities among treatment levels. For adults, tank of origin was included as a random effect. We used principal coordinate analyses on unweighted UniFrac distances to visualize similarities of bacterial community membership across water treatments. Unweighted scores represent bacterial community membership, which is based on the presence or absence of bacterial taxa, whereas weighted scores represent bacterial community structure, which also takes into account relative abundance of bacterial taxa.

To compare relative abundances of bacterial taxa across groups, we first removed any phyla that were present in <25% of samples. Given that the gut bacterial community is largely restructured over the course of metamorphosis[19], we compared relative abundances of bacteria in juveniles and adult frogs separately. Relative abundances (arcsine square root transformed)[39, 40] of bacterial phyla in juveniles and adults were analyzed in JMP (version 12) using ANOVAs with water treatment as an independent variable and, for adults, with tank as a random effect. For all analyses, $P$-values were corrected using the false discovery rate correction for multiple comparisons. We used GLMs (for juveniles) and GLMMs (for adults with tank as a random effect) to compare the relationship between relative abundance of phyla and genera and parasite establishment.

To determine the overall relationship among water treatment, juvenile and adult bacterial phylogenetic diversity, and adult parasite susceptibility, we employed SEM and factor analyses. SEM is a statistical technique based on the analysis of variance–covariance matrices[41], which can test web-like hypotheses because variables can serve as both independent and dependent variables. The type of analysis chosen for a candidate model was dependent on whether juvenile skin and gut microbiota were combined as a latent variable (SEM; which combines factor analysis with path analysis) or considered separate variables (path analysis only). Analyses were conducted using the lavaan package in RStudio. SEMs were only conducted on tanks for which we had all the measured variables: juvenile bacterial diversity (gut and skin), adult gut bacterial diversity, and infection data. Water treatment was either categorized as control (pond water) or experimental (sterile pond water, short-term antibiotic exposure, or long-term antibiotic exposure) for the SEM. We tested eight a priori hypothesized path models that removed various sets of paths from the starting full model (Supplementary Fig. 1, Supplementary Table 2). These eight models were ranked using AIC.

**Data availability**. The 16S recombinant DNA sequences have been deposited in the BioProject database under accession code PRJNA342830. The authors declare that all other relevant data supporting the findings of the study can be found on FigShare (doi: 10.6084/m9.figshare.5044825).

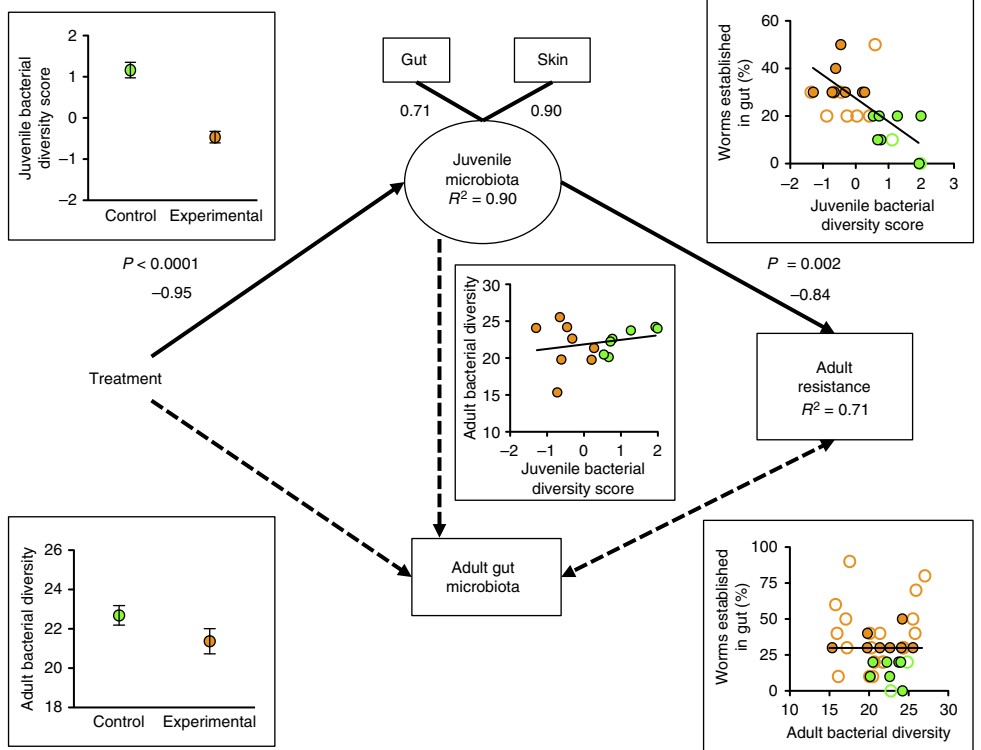

**Fig. 5** Early-life disruption of juvenile skin and gut microbiota predicts later-life resistance to infection. Paths included in the best structural equation model (SEM) based on Akaike information criterion are depicted with *solid lines*; paths not included in the best model are depicted with *dotted lines*. The best model shows that the experimental water treatment reduced bacterial phylogenetic diversity in juveniles, and in turn, bacterial diversity in juveniles negatively predicted worm establishment in the guts of adults. The best model combines gut and skin bacterial diversity of juveniles into a latent variable (because they are positively correlated; GLM, $\chi^2 = 31.95$, df = 1, $P < 0.0001$) and therefore we use a factor score to present these data; numbers below the "skin" and "gut" boxes are factor loadings. In contrast, treatment did not affect bacterial diversity of adults, bacterial diversity of juveniles was not related significantly to bacterial diversity of adults, and bacterial diversity of adults was not related significantly to adult resistance to infection. The SEM only included tanks for which there was a complete set of samples for skin and gut bacteria of juveniles, gut bacteria of adults, and infection data ($n = 15$; *solid points*); *clear points* represent the remaining data. Control pond water treatment is represented in *green* and experimental water treatments (sterile pond water, sterile pond water plus antibiotics) are represented in *orange*. Individual regressions based on the entire available data set provide similar results as the SEM on the subset of complete data and thus we conservatively provide the results from only the subset. Error bars indicate the s.e.m. *P*-values and standardized coefficients from the best model are shown next to each path. $R^2$-values indicate total variance explained by predictor variables in the top model. The $\chi^2$ of the best model was 0.38 (df = 2, $P = 0.83$), indicating that the model was a good fit to the data. This model accounted for 92% of the model weight

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

## Acknowledgements

We thank Lauren Shea, Virginia Caponera, Amanda Dubour, Sophia Estrada, Sahara Peters, and Shaun Sehgal for assistance with the study and Doug Woodhams and Whitney Holden for helpful discussions on experimental design. We also thank Elisabeth Bik and two anonymous reviewers for comments that improved the manuscript. This research was funded by grants from the British Ecological Society (5599–6643) to S.A.K., and National Science Foundation (EF-1241889), National Institute of Health (R01GM109499 and R01TW010286), United States Department of Agriculture (NRI 2006-01370,2009-35102-0543), and Environmental Protection Agency (CAREER 83518801) to J.R.R.

## Author contributions

Conception and experimental design: S.A.K. and J.R.R. Methodology and data acquisition: S.A.K. and C.L.W. Analysis and interpretation of data: S.A.K., K.D.K., and J.R.R. Manuscript writing: S.A.K. and J.R.R.

## Additional information

**Competing interests:** The authors declare no competing financial interests.

