## [Peer Review File · Nature Communications]

Reviewers' comments:

Reviewer #1 (Remarks to the Author):

In this manuscript, Knutie et al. reared tadpoles in four different types of water that varied in sterility and antibiotic treatment. Abx treatment of juveniles resulted in decreased bacterial diversity and increased susceptibility for parasite worm establishment in adult frogs. Adult bacterial diversity in adult frogs was not correlated to parasite establishments in adults.

This paper is well written and figures are clear. Most of my comments are minor and can be easily addressed by the authors.

General comments

1. The first 2 paragraphs of the introduction did not seem to flow very well. Line 31-37 address microbiota diversity just before parasite infection, while L38 and following talk about diversity early in life affecting later in life. However, these 2 different viewpoints were not immediately different – it was only after reading it twice that I saw the contradiction. Is there a way to better emphasize the 2 different topics here? Maybe by starting L38 with “In contrast to this immediate effect” ? Also, in L32, I was not sure if “ability of the host to reduce infections” was worded correctly, especially since this is the first line of the manuscript. Would it not be better to state a more neutral “ability of the parasite to cause infection/establish” without indicating yet if this is a direct effect of the microbiota diversity or indirectly by a change on the host’s side (e.g. by changing the immune system)? The next two lines introduce both possibilities, so it might be better to remain a bit more neutral in the first sentence.
2. Could the authors provide a bit more information on Aplectana worm infection of these frogs, maybe around line 68? How does a frog typically acquire these? Through the skin as suggested in L68 or by drinking contaminated water? And where do they typically reside after infection? In the gut? How were these worms collected, by dissecting infected frogs (L170)? From the later text (e.g. around L97) it appears that “worm penetration” is something different than “worm establishment” but without some more background on the mechanisms and routes of parasite acquisition and infection, the interpretation of the results shown in figures 3a and 3b was a bit hard to follow.
3. The number of animals used in the adult analysis (day 160 or 180) was not clear, although the numbers were indicated in Figure 2a-c. The experiment started with 80 juveniles per treatment group, and 4 animals per tank, if I understood correctly. Ten juveniles per treatment group were used in the juvenile gut/skin bacterial diversity analysis. That would leave roughly 70 animals for the adult parasite exposure and necropsy, but the numbers in Figure 2c are much lower. Did the authors only select ~10 per treatment group for the parasite infection, and was this done per tank? Maybe this number could be added around line 169? Did more animals die during metamorphosis or later in one of the treatment groups?
4. I did not understand how Supplemental Figure 2a and b could have been generated. Juveniles were sacrificed to determine bacterial diversity; worm infection was done in adults. So there are no animals who delivered the data plotted on the 2 respective axes. E.g. the animal who had a juvenile bacterial diversity of 44 is not the same as the adult animal with 0% worms. How did the authors pair these datapoints? Were they paired per tank? This needs to be clarified.
5. The Fusobacteria findings would be easier to understand if the authors could provide a (supplemental) figure showing the relative abundance of this taxon and other taxa (as mentioned in the supplemental Results) in the 4 different treatment groups in juveniles and adults. This could be e.g. a stacked column/bar graph. Also, with “Fusobacteria”, do the authors mean the phylum or the genus? This was not clear from the main text.
6. Assuming Fusobacteria meant the phylum (based on the description of the phylum-level analysis in the supplemental results), did the authors test for significant differences on other taxonomic levels, e.g. genus? Where were specific genera associated with an increased susceptibility for parasite infection?

Minor comments

7. L65. "16S rRNA gene"

8. L76-78 were outside my area of expertise; this was very technical. Some more explanation, e.g., a couple of lines in the supplemental methods/results might be helpful for many readers.

9. L93. I was not sure if the long term antibiotics-treatment samples (triangle) were very different from the other samples. They appear to be not clustering separately. Could the authors clarify? Maybe they separated in one of the other coordinates, but I did not see it here. I do agree with the statement in L93-96, though.

10. L145. "From here on" is not completely correct as this was also stated in L85 ☺ Also maybe include "each" to indicate that each tank received 4 tadpoles. What is meant by the Gosner stage – maybe the authors can include a reference here? Does the number refer to the Gosner stage or is this something else (weight? Number of animals?).

11. L193. What is meant by "OTUs were classified using UCLUST" ? Do the authors mean taxonomic classification using a reference set (which one?), or clustering into phylogenetic groups (which would be confusing since they were already clustered into OTUs)

12. L195. What is meant by the "blank samples"? Are these extraction controls, or samples from the sterilized water? Please indicate at what point in the experiments these samples were collected.

13. L216 and L367. This should read "Principal Coordinate Analysis". Please include a reference in L216.

14. Figure 1. This figure was very helpful. But - it would be even better if the authors could add the number of animals used per analysis (e.g. 10 per treatment group on day 30). How many frogs per treatment group were on average used for the parasite infection and analysis on day 160/180?

15. Figure 2. The a,b,c above the mean values in panels a through c were potentially confusing because these letters are the same as the panel designations. The authors can choose to ignore this remark; it's very minor. In L370, should "water exposure" not be "water sterilization" or something similar? All tadpoles were exposed to water. In panels d-f, what does the ellipse/circle indicate? Also, maybe the authors could clarify that each datapoint in panels d-f represents an individual animal

16. Figure 4. Less diverse bacterial communities in juveniles are associated with higher % worms in the adult gut. Is the statement "experimental reduction of bacterial phylogenetic diversity in juveniles negatively predicts worm establishment in the guts of adults" then correct? Should this be "positively" ? This might be my lack of understanding the SEM analysis, but maybe the authors could clarify.

17. Suppl Figure 2. How is bacterial diversity defined here? Is this number of observed species or Faith's diversity? Also, how could panel a and b be calculated? In order to determine the bacterial diversity, the juveniles were killed, but the worm experiments were done in the adults. There is therefore no animal that could have delivered to both datapoints?

Signed: Elisabeth Bik, Stanford University

Reviewer #2 (Remarks to the Author):

Knutie et al. have performed a fascinating study on how early life disruption of microbiota can influence infection later in life. This is an important topic of broad disciplinary interest. Increasing evidence suggests that broad-spectrum antibiotics can negatively impact human health, especially when applied early in development. Here, an amphibian model is used to empirically test hypotheses about microbiota reduction and infection risk. That said, bacterial abundance was not measured in this study, and "disruption" refers only to the relative abundance of bacterial taxa and

overall community composition. Is it possible that under experimental conditions of sterile water and long-term antibiotics, some bacteria are freed from competition and thus bloom? I would like to see some discussion on the importance of bacterial density/abundance vs. community composition.

There are also several pieces of data missing from this short report, that could easily be included in the supplement. For example, of the 320 tadpoles used in the study, how many were sampled or survived along the way? No survival results from the treatments are mentioned, but feeding with sterilized food is likely to have some impact on nutrition and thus survival. Could the greater infection risk later in life from these "disrupted" or bacterially-depleted frogs be linked with differences in size at metamorphosis, or other general health features of the frogs related to the rearing conditions, but not directly caused by the lack of microbiota? While the phenomenon is certainly of great interest, no immune mechanisms are quantified in this study. It would have been nice to see some mention (beyond unpublished data line 123) of what the anti-Aplectana defenses in amphibians are, and some quantification of whether the treatments impacted those defenses. Was there an effect on immunity, not just infectivity? Would this have the potential to increase disease risk in general, or was it specific to parasitic gut worms? Is the microbiota itself a defense against the worms?

Why was adult skin microbiota not also measured? This may be more important than adult gut microbiota in determining penetration of the skin and infectivity.

Line 70 – edit to "which occurred in adults five months after"

Line 112 – While Fusobacteria in juvenile skin went down with antibiotics, Actinobacteria was 20x higher (Supplemental data). It is difficult to interpret the taxonomic data from this study because no taxa plots showing relative abundance in the various treatments, life-stages, and tissues sampled are provided. These should be added to the supplement at a minimum. Perhaps general bacterial primers could be used in a qPCR using the original DNA extractions. This would help determine whether an Actinobacteria bloom was caused by reducing most of the bacteria with antibiotics.

136 change to "as a consequence"

152 – Antibiotics were added once, or at each water change? If at each water change, does this mean 4 times, or once a week until sampling at day 30?

164 – The 70? juveniles in each treatment were allowed to metamorphose? Were there differences among treatments in time to metamorphosis or survival, or size at metamorphosis?

177 – How did you determine if the missing Aplectana had actually penetrated the frog skin vs. just stuck to the frogs before removal from the exposure chamber? Were the frogs rinsed?

195 – A list of the blank sample "contaminant" OTUs would be helpful. Where did these contaminants come from?

199 – Why 1900 sequences per sample?

Reviewer #3 (Remarks to the Author):

The manuscript presents a compelling case for the disruption of early life microbiota affecting adult parasite resistance. For the specific interaction studied, limiting exposure of live microbes to early stage tadpoles decreases nematode resistance of adult frogs. As such, the study is novel and of potentially broad interest to host-parasite ecology/ ecological immunity and beyond.

The experiments were well conducted and are clearly and concisely described. The introduction and discussion serve well to put the study into a broader context.

While I consider the conclusions to be mostly sound, the following comments may improve the manuscript:

-Line 55ff: This statement seems to suggest that only the bacterial microbiota is experimentally manipulated. However, given the design of the study, the observed effect could come from any organism in the pond water that is inactivated by autoclaving, including also fungi, protists or viruses. Maybe there are reasons to assume that bacteria are the most likely cause of the effect, but since the other microbial groups were not looked at, I feel a caveat should be added.

-The manuscript could be improved by a better description of the taxonomic variation in the microbiota of the skin and gut in the different treatments. I would appreciate a figure in the supplementary material that illustrates the relative abundance of the main bacterial OTUs in bar charts for each individual organised by treatment. If indeed Fusobacteria are the main factor in which the microbiota between the control and treatment groups differ, this would become visually clearer through this. In line 193f the authors mention that a phylogenetic tree of the OTUs was generated, including this in the supplementary would also help the reader to understand the microbial diversity in the experiment in a better way. In addition, it would help the reader and future researchers in this area if representative sequences for the OTUs and potentially the complete sequencing data set would be made publicly available. At the moment the information obtainable by the reader is restricted to descriptions of statistical test results on the phylum level in the supplementary material.

-Line 174ff: By the description of the assay, it's not clear to me how a penetration of the frogs by the worms can be assumed just by the absence of the worms in the cup? Is it not equally possible that worms were on the surface of the frogs (therefore absent from the cup), but failed to penetrate? In this case, it would seem misleading to suggest that actual successful penetration was measured in the assay.

Reviewers' comments (italics) and authors' responses (bold):

Reviewer #1 (Remarks to the Author):

In this manuscript, Knutie et al. reared tadpoles in four different types of water that varied in sterility and antibiotic treatment. Ab treatment of juveniles resulted in decreased bacterial diversity and increased susceptibility for parasite worm establishment in adult frogs. Adult bacterial diversity in adult frogs was not correlated to parasite establishments in adults.

This paper is well written and figures are clear. Most of my comments are minor and can be easily addressed by the authors.

General comments

1. The first 2 paragraphs of the introduction did not seem to flow very well. Line 31-37 address microbiota diversity just before parasite infection, while L38 and following talk about diversity early in life affecting later in life. However, these 2 different viewpoints were not immediately different – it was only after reading it twice that I saw the contradiction. Is there a way to better emphasize the 2 different topics here? Maybe by starting L38 with “In contrast to this immediate effect” ?

We changed the wording to “In contrast to the direct effects...” on line 35.

Also, in L32, I was not sure if “ability of the host to reduce infections” was worded correctly, especially since this is the first line of the manuscript. Would it not be better to state a more neutral “ability of the parasite to cause infection/establish” without indicating yet if this is a direct effect of the microbiota diversity or indirectly by a change on the host’s side (e.g. by changing the immune system)? The next two lines introduce both possibilities, so it might be better to remain a bit more neutral in the first sentence.

We changed the wording of the first sentence of the introduction to: “A disruption of the normal microbiota of hosts just before parasite exposure has been shown to increase infection risk” on lines 31-32.

2. Could the authors provide a bit more information on Aplectana worm infection of these frogs, maybe around line 68? How does a frog typically acquire these? Through the skin as suggested in L68 or by drinking contaminated water? And where do they typically reside after infection? In the gut? How where these worms collected, by dissecting infected frogs (L170)? From the later text (e.g. around L97) it appears that “worm penetration” is something different than “worm establishment” but without some more background on the mechanisms and routes of parasite acquisition and infection, the interpretation of the results shown in figures 3a and 3b was a bit hard to follow.

Since submitting the paper, Aplectana sp. has been identified to species. We added the following information about Aplectana hamatospicula (Ascaridida: Cosmocercidae) to lines 72-76: “A. hamatospicula has a direct life cycle: juvenile larvae penetrate frog skin and then, in approximately 3 weeks, establish, mature and reproduce in the gastrointestinal tract. Worm eggs and larvae (they are ovoviviparous) are defecated by frogs, and after approximately a week of development, juveniles can infect the next host.” We state in the methods that the worms “were collected from naturally-parasitized adults” on line 234.

3. The number of animals used in the adult analysis (day 160 or 180) was not clear, although the numbers were indicated in Figure 2a-c. The experiment started with 80 juveniles per treatment group, and 4 animals per tank, if I understood correctly. Ten juveniles per treatment group were used in the juvenile gut/skin bacterial diversity analysis. That would leave roughly 70 animals for the adult parasite exposure and necropsy, but the numbers in Figure 2c are much lower. Did the authors only select ~10 per treatment group for the parasite infection, and was this done per tank? Maybe this number could be added around line 169? Did more animals die during metamorphosis or later in one of the treatment groups?

We added sample sizes (individuals per tub) for each time point in the experiment in figure 1.

4. I did not understand how Supplemental Figure 2a and b could have been generated. Juveniles were sacrificed to determine bacterial diversity; worm infection was done in adults. So there are no animals who delivered the data plotted on the 2 respective axes. E.g. the animal who had a juvenile bacterial diversity of 44 is not the same as the adult animal with 0% worms. How did the authors pair these datapoints? Were they paired per tank? This needs to be clarified.

Yes, they were paired per tank as suggested. Sorry for the confusion. Juvenile and adult samples were collected from different individuals within the same replicate (tank) because tadpole sampling required destructive sampling. Thus individuals from the same tank were paired for analyses. This information is now included in the legend for Supplemental Figure 2 and the statistical methods on lines 288-291.

5. The *Fusobacteria* findings would be easier to understand if the authors could provide a (supplemental) figure showing the relative abundance of this taxon and other taxa (as mentioned in the supplemental Results) in the 4 different treatment groups in juveniles and adults. This could be e.g. a stacked column/bar graph.

We added stacked column bar graphs of relative abundance of phyla for tadpole skin and guts and adult guts (Figure 3).

Also, with “*Fusobacteria*”, do the authors mean the phylum or the genus? This was not clear from the main text.

We now identify that *Fusobacteria* is a phylum on line 111.

6. Assuming *Fusobacteria* meant the phylum (based on the description of the phylum-level analysis in the supplemental results), did the authors test for significant differences on other taxonomic levels, e.g. genus? Were there specific genera associated with an increased susceptibility for parasite infection?

Great point. Within phylum *Fusobacteria*, we found that the relative abundance of bacteria from the genus *Cetobacterium* of the guts and skin of juvenile frogs negatively predicted the number of worms in the guts of adults. We now include these results in the main text (lines 140-143) as well as in an inclusive table (S4) of the genera that were associated with host resistance in the supplemental results.

Minor comments

7. L65. “16S rRNA gene”

We changed “16S rRNA” to “16S rRNA gene” on line 64.

8. L76-78 were outside my area of expertise; this was very technical. Some more explanation, e.g., a couple of lines in the supplemental methods/results might be helpful for many readers.

We added more information about SEM in the methods on lines 318-323. “SEM is a statistical technique based on the analysis of variance–covariance matrices (Grace 2006), which can test web-like hypotheses because variables can serve as both independent and dependent variables. The type of analysis chosen for a candidate model was dependent on whether juvenile skin and gut microbiota were combined as a latent variable (SEM; which combines factor analysis with path analysis) or considered separate variables (path analysis only).”

Grace, J. Structural equation modeling and natural systems. (Cambridge University Press, 2006).

9. L93. I was not sure if the long term antibiotics-treatment samples (triangle) were very different from the other samples. They appear to be not clustering separately. Could the authors clarify? Maybe they separated in one of the other coordinates, but I did not see it here. I do agree with the statement in L93-96, though.

This is a very astute observation. The separation of the long-term antibiotic treatment vs. other treatments in adults actually occurs on the 2nd and 3rd axes. Thus, we chose to change figure 2f to include axes 2 and 3 instead of 1 and 2.

10. L145. “From here on” is not completely correct as this was also stated in L85 Also maybe include “each” to indicate that each tank received 4 tadpoles. What is meant by the Gosner stage – maybe the authors can include a reference here? Does the number refer to the Gosner stage or is this something else (weight? Number of animals?).

Lines 205-208 now read: “Eighty 4L tanks randomly received four *O. septentrionalis* tadpoles (juveniles) each, which were collected from pools at the University of South Florida (USF) Botanical Gardens. The mean (\pm SE) Gosner stage for juveniles at the beginning of the experiment was 32.10 (\pm 1.10); Gosner stage represents the developmental stage of advancement towards metamorphosis in tadpoles³³.”

11. L193. What is meant by “OTUs were classified using UCLUST” ? Do the authors mean taxonomic classification using a reference set (which one?), or clustering into phylogenetic groups (which would be confusing since they were already clustered into OTUs?)

We have expanded this sentence to now read “OTUs were classified taxonomically using UCLUST³⁷ with the reference Greengenes database³⁴.” on lines 268-269.

12. L195. What is meant by the “blank samples”? Are these extraction controls, or samples from the sterilized water? Please indicate at what point in the experiments these samples were collected.

We also extracted and sequenced “blank” samples, which were collected using sham-necropsies and sham-extractions (i.e. without an experimental sample) to control for methodological contamination (Salter et al. 2014). This information is now included on lines 256-259.

Salter, S. J. *et al.* Reagent and laboratory contamination can critically impact sequence-

based microbiome analyses. *BMC Biol.* **12**, 87 (2014).

13. L216 and L367. This should read “Principal Coordinate Analysis”. Please include a reference in L216.

We changed “Principle” to “Principal” on lines 301 and 453.

14. Figure 1. This figure was very helpful. But - it would be even better if the authors could add the number of animals used per analysis (e.g. 10 per treatment group on day 30). How many frogs per treatment group were on average used for the parasite infection and analysis on day 160/180?

Great suggestion. We added sample sizes (individuals per tub) for each time point in the experiment in figure 1.

15. Figure 2. The a,b,c above the mean values in panels a through c were potentially confusing because these letters are the same as the panel designations. The authors can choose to ignore this remark; it's very minor.

We changed lettering above mean values to capital letters to distinguish from the panel letters.

In L370, should “water exposure” not be “water sterilization” or something similar? All tadpoles were exposed to water.

We changed “water exposure” to “water treatment exposure” in the figure legend on line 457-458.

In panels d-f, what does the ellipse/circle indicate? Also, maybe the authors could clarify that each datapoint in panels d-f represents an individual animal

The circles, which relate to the vector overlay, represent a unit circle (radius = 1) to demonstrate the direction and correlational strength of vector A. The relative size and position of the origin is arbitrary with respect to the underlying plot. We added a description of the circle on lines 455-457.

16. Figure 4. Less diverse bacterial communities in juveniles are associated with higher % worms in the adult gut. Is the statement “experimental reduction of bacterial phylogenetic diversity in juveniles negatively predicts worm establishment in the guts of adults” then correct? Should this be “positively” ? This might be my lack of understanding the SEM analysis, but maybe the authors could clarify.

We changed the figure legend to clarify that “The best model shows that the experiment water treatment reduced bacterial phylogenetic diversity in juveniles and bacterial diversity in juveniles negatively predicted worm establishment in the guts of adults.” On lines 493-495.

17. Suppl Figure 2. How is bacterial diversity defined here? Is this number of observed species or Faith's diversity? Also, how could panel a and b be calculated? In order to determine the bacterial diversity, the juveniles were killed, but the worm experiments were done in the adults. There is therefore no animal that could have delivered to both datapoints?

We changed “bacterial diversity” to “bacterial phylogenetic diversity” in the legend of

Supplemental Figure 2). We also now clarify how we linked tadpole and adult variables by adding the following to the legend: “Tadpole and adult samples were collected from different individuals within the same replicate (tank) because tadpole sampling required destructive sampling. Juvenile and adult samples from the same tank were then paired for analyses.”

Signed: Elisabeth Bik, Stanford University

Reviewer #2 (Remarks to the Author):

Knutie et al. have performed a fascinating study on how early life disruption of microbiota can influence infection later in life. This is an important topic of broad disciplinary interest. Increasing evidence suggests that broad-spectrum antibiotics can negatively impact human health, especially when applied early in development. Here, an amphibian model is used to empirically test hypotheses about microbiota reduction and infection risk. That said, bacterial abundance was not measured in this study, and “disruption” refers only to the relative abundance of bacterial taxa and overall community composition. *Is it possible that under experimental conditions of sterile water and long-term antibiotics, some bacteria are freed from competition and thus bloom? I would like to see some discussion on the importance of bacterial density/abundance vs. community composition.*

Great point and suggestion. Absolute abundance of bacterial taxa may also play a role in infection risk but was not quantified in our study. A common method to quantify absolute abundance is quantitative PCR (qPCR) to determine the copies of 16S rRNA per gram of gut contents. However, we extracted bacterial DNA from the entire gut (intestines and contents) to maximize biomass recovery and therefore did not specifically weigh the gut contents. Weighing only the contents (as opposed to total gut mass) for qPCR is important because the mass of the contents depends on several variables, such as when the animal last ate or defecated. Future studies could include absolute abundance data by removing, weighing, and extracting bacterial DNA from only the contents of the intestines. We now include this caveat in the discussion on lines 176-182.

There are also several pieces of data missing from this short report, that could easily be included in the supplement. For example, of the 320 tadpoles used in the study, how many were sampled or survived along the way? No survival results from the treatments are mentioned, but feeding with sterilized food is likely to have some impact on nutrition and thus survival. Could the greater infection risk later in life from these “disrupted” or bacterially-depleted frogs be linked with differences in size at metamorphosis, or other general health features of the frogs related to the rearing conditions, but not directly caused by the lack of microbiota?

To clarify, juveniles from all treatments were fed sterilized food, which is stated on lines 62 and 220. However, we agree that the effect of treatment on host health may affect infection risk. We now include results on the effect of water treatment on mass at metamorphosis, time to metamorphosis, and survival and the effect of these variables on infection risk. Juveniles from the long-term antibiotic treatment had lower survival and took longer to metamorphose than the other treatments. Juvenile mass at metamorphosis did not differ significantly across treatments. In contrast, juvenile water treatment affected adult mass but not survival. Juvenile mass at metamorphosis, time to metamorphosis, and adult mass

did not affect infection risk. Lines 93-104.

While the phenomenon is certainly of great interest, no immune mechanisms are quantified in this study. It would have been nice to see some mention (beyond unpublished data line 123) of what the anti-Aplectana defenses in amphibians are, and some quantification of whether the treatments impacted those defenses. Was there an effect on immunity, not just infectivity? Would this have the potential to increase disease risk in general, or was it specific to parasitic gut worms? Is the microbiota itself a defense against the worms?

Indeed, the early-life microbiota affected later-life infection risk, which suggests that the early-life microbiota affected resistance mechanisms that were lasting into adulthood. We did not find evidence that the adult microbiota directly affects worms, which is stated in the main text and supplemental results. Tissue, such as plasma, was not collected for this study to quantify the effect of treatment on specific immune measures, such as IgY antibodies. However, in a recent published study, we demonstrate that frogs produce an acquired IgY antibody response to *Aplectana* and this paper is now cited (Knutie et al. 2017)(line 160-161). Identifying the other potential resistance mechanism(s) requires characterizing host gene expression (e.g. using RNA-seq) of candidate immune genes in both tadpoles and adults (lines 162-164), which is very expensive, but feasible in the future with funding. Knutie, S. A., Wilkinson, C. L., Wu, Q. C., Ortega, C. N. & Rohr, J. R. Host resistance and tolerance of parasitic gut worms depend on resource availability. *Oecologia* 1–10 (2017).

Why was adult skin microbiota not also measured? This may be more important than adult gut microbiota in determining penetration of the skin and infectivity.

We did not find a significant effect of juvenile water treatment on worm penetration in adults (lines 119-120) and most frogs were penetrated by at least 75% of the worms (Fig. 4). Thus, we predicted that the role of skin bacteria played a minor role, if any, in resistance to worm penetration. Given that the skin microbiota of tadpoles was not predictive of worm penetration success, we had no reason to think it would be important for the same worms as adults. As a result, adult skin samples were not collected and thus not included in the study.

Line 70 – edit to “which occurred in adults five months after”

We changed “which occurred five months after” to “which occurred in adults five months after” on lines 77.

*Line 112 – While *Fusobacteria* in juvenile skin went down with antibiotics, *Actinobacteria* was 20x higher (Supplemental data). It is difficult to interpret the taxonomic data from this study because no taxa plots showing relative abundance in the various treatments, life-stages, and tissues sampled are provided. These should be added to the supplement at a minimum.*

Great point! We added stacked column bar graphs of relative abundance of phyla for tadpole skin and guts and adult guts (Figure 3).

*Perhaps general bacterial primers could be used in a qPCR using the original DNA extractions. This would help determine whether an *Actinobacteria* bloom was caused by reducing most of the bacteria with antibiotics.*

As stated above, absolute abundance of bacterial taxa may also play a role in infection risk

but was not quantified in our study. A common method to quantify absolute abundance is quantitative PCR (qPCR) to determine the copies of 16S rRNA per gram of gut contents. However, we extracted bacterial DNA from the entire gut (intestines and contents) to maximize biomass recovery and therefore did not specifically weigh the gut contents. Weighing only the contents (as opposed to total gut mass) for qPCR is important because the mass of the contents depends on several variables, such as when the animal last ate or defecated. Future studies could include absolute abundance data by removing, weighing, and extracting bacterial DNA from only the contents of the intestines. See lines 176-182.

136 change to “as a consequence”

We changed “as consequence” to “as a consequence” on line 197.

152 – Antibiotics were added once, or at each water change? If at each water change, does this mean 4 times, or once a week until sampling at day 30?

To clarify our methods, we added the following statement on lines 215-216: “For the long-term antibiotic treatment, the antibiotic cocktail was added to the tank during each weekly water change.”

164 – The 70? juveniles in each treatment were allowed to metamorphose? Were there differences among treatments in time to metamorphosis or survival, or size at metamorphosis?

We added sample sizes (individuals per tub) for each time point in the experiment in figure 1. We now include results on the effect of water treatment on mass at metamorphosis, time to metamorphosis, and survival and the effect of these variables on infection risk. Juveniles from the long-term antibiotic treatment had lower survival and took longer to metamorphose than the other treatments. Juvenile mass at metamorphosis did not differ significantly across treatments. In contrast, juvenile water treatment affected adult mass but not survival. Juvenile mass at metamorphosis, time to metamorphosis, and adult mass did not affect infection risk. Lines 93-104.

177 – How did you determine if the missing Aplectana had actually penetrated the frog skin vs. just stuck to the frogs before removal from the exposure chamber? Were the frogs rinsed?

In a separate experiment, now described on lines 247-252, we determined whether juvenile worms could be observed on the surface of the frog’s skin without penetrating the skin successfully after parasite exposure. We exposed five additional frogs to juvenile worms using the methods described in the main text. After 24 hours, we anesthetized frogs and scanned the surface of the skin for visible worms that did not penetrate it successfully. We did not observe any juvenile worms on the surface of the skin, which suggests that if worms were not found in the petri dish, they penetrated the host successfully.

195 – A list of the blank sample “contaminant” OTUs would be helpful. Where did these contaminants come from?

Blank samples were collected using sham-necropsies and sham-extractions (i.e. without an experimental sample) to control for methodological contamination (lines 256-259). For example, there is microbial DNA present in DNA extraction kits that can confound the

study. One technique to prevent contamination is to remove any OTUs present in ‘blank’ extractions, which is proposed in Salter et al. 2014 (line 271). We did not list the contaminant OTUs as they were largely similar to those presented in Salter et al (line 272). The information above is now stated in the main text.

Salter, S. J. *et al.* Reagent and laboratory contamination can critically impact sequence-based microbiome analyses. *BMC Biol.* 12, 87 (2014).

199 – Why 1900 sequences per sample?

This number was based off of the number of sequences returned from each sample. Given that some samples only had ~1900 sequences, we could only rarefy all samples to this number. This is a common technique when analyzing microbiome data. We have amended the end of this sentence to now read “the latter of which measures the cumulative branch lengths from randomly sampling 1900 sequences from each sample (the minimum number of sequences returned from each sample).” On line 276.

Reviewer #3 (Remarks to the Author):

The manuscript presents a compelling case for the disruption of early life microbiota affecting adult parasite resistance. For the specific interaction studied, limiting exposure of live microbes to early stage tadpoles decreases nematode resistance of adult frogs. As such, the study is novel and of potentially broad interest to host-parasite ecology/ ecological immunity and beyond.

The experiments were well conducted and are clearly and concisely described. The introduction and discussion serve well to put the study into a broader context.

While I consider the conclusions to be mostly sound, the following comments may improve the manuscript:

-Line 55ff: This statement seems to suggest that only the bacterial microbiota is experimentally manipulated. However, given the design of the study, the observed effect could come from any organism in the pond water that is inactivated by autoclaving, including also fungi, protists or viruses. Maybe there are reasons to assume that bacteria are the most likely cause of the effect, but since the other microbial groups were not looked at, I feel a caveat should be added.

We changed “microbial” to “bacterial” throughout the manuscript to clarify that we focused on bacteria rather than the entire microbial community. Early-life exposure to other microbes may affect later-life infection risk as well, which is certainly an avenue of research that should be explored in the future. A caveat is now added to the discussion on lines 182-185.

-The manuscript could be improved by a better description of the taxonomic variation in the microbiota of the skin and gut in the different treatments. I would appreciate a figure in the supplementary material that illustrates the relative abundance of the main bacterial OTUs in bar charts for each individual organised by treatment. If indeed Fusobacteria are the main factor in which the microbiota between the control and treatment groups differ, this would become visually clearer through this. In line 193f the authors mention that a phylogenetic tree of the OTUs was generated, including this in the supplementary would also help the reader to understand the microbial diversity in the experiment in a better way.

We added stacked column bar graphs of relative abundance of phyla for tadpole skin and guts and adult guts (Figure 3). The tree file generated is for 1900 OTUs identified in the data set, which is a very large tree, and not the best way to look for phylogenetic signals in our experiments. Moreover, the labeling at the tips of the trees contain OTU numbers generated by our specific analysis, which would not be useful to another researcher. Our sequences have been uploaded to NCBI database for researchers who wish to conduct their own analyses.

In addition, it would help the reader and future researchers in this area if representative sequences for the OTUs and potentially the complete sequencing data set would be made publicly available. At the moment the information obtainable by the reader is restricted to descriptions of statistical test results on the phylum level in the supplementary material.

Great point! We added the following information to the methods: “All sequences were deposited in the Sequence Read Archive under accession PRJNA342830.” On lines 280-281.

-Line 174ff: By the description of the assay, it's not clear to me how a penetration of the frogs by the worms can be assumed just by the absence of the worms in the cup? Is it not equally possible that worms were on the surface of the frogs (therefore absent from the cup), but failed to penetrate? In this case, it would seem misleading to suggest that actual successful penetration was measured in the assay.

In a separate experiment, now described on lines 247-252, we determined whether juvenile worms could be observed on the surface of the frog's skin without penetrating the skin successfully after parasite exposure. We exposed five additional frogs to juvenile worms using the methods described in the main text. After 24 hours, we anesthetized frogs and scanned the surface of the skin for visible worms that did not penetrate it successfully. We did not observe any juvenile worms on the surface of the skin, which suggests that if worms were not found in the petri dish, they penetrated the host successfully.

Reviewers' comments:

Reviewer #1 (Remarks to the Author):

The authors have responded to all comments made by the reviewers, and I have no further comments.

I congratulate the authors for a job well done and hope to see this paper published in Nature Communications.

Reviewer #2 (Remarks to the Author):

The main conclusion is this:

"Thus, we demonstrate that an early-life disruption of the microbiota had lasting reductions on host resistance to infections, which was likely mediated by its effects on immune system development."

I remain unconvinced of this conclusion.

Yes, the antibiotic treatment reduced bacteria in tadpoles.

It also reduced growth rate and probably immune system development.

So it would be just as accurate to say that reduced growth rate as larva, or antibiotic toxicity, had lasting reductions on host resistance to infections, and thus preventing early life disruption of growth, or preventing antibiotic toxicity, could affect long-term health.

There are no treatments showing reduced microbiota independent of growth rate caused the same increase in infection as adults.

There are no experiments showing how the gut microbiota of tadpoles are involved mechanistically in infection susceptibility as adults.

Thus, I cannot accept the conclusion without a major caveat in the abstract, introduction, and discussion wherever this result is reported.

A trade off between growth and (immune) development is widely known for developing amphibians.

This study may be another demonstration of this, and have nothing to do with the microbiota.

I appreciate the new first section of the results that gives details on the reduced growth rate of antibiotic treated tadpoles,

reduced growth rate of juveniles that were previously antibiotic treated,

and lower survival of antibiotic treated tadpoles.

I believe these results are key to understanding and interpreting this study:

"Effect of water treatment on host health. Water treatment significantly affected juvenile survival (Coxme, $\chi^2 = 16.69$, $df = 3$, $P < 0.001$; Fig. 1; Supplemental Table 2) and time to metamorphosis (GLMM, $\chi^2 = 120.85$, $df = 3$, $P < 0.0001$). Juveniles from the long-term antibiotic treatment took twice as long to metamorphose and had lower survival compared to juveniles from the other treatments (sequential Bonferroni post-hoc multiple comparison test: $P < 0.05$ for long-term antibiotic frogs compared to all other treatments). ... juvenile water treatment significantly affected adult mass when measured two months after metamorphosis (GLMM, $\chi^2 = 19.44$, $df = 3$, $P < 0.001$); adults exposed to long-term antibiotics weighed less than frogs from the other treatments ($P < 0.05$ for long-term antibiotic frogs compared to all other treatments). "

The lower survival is an indication that the animals are not in the best health, but this may or may not be because of a reduced microbiota as tadpoles.

At high densities, or under other stressful conditions, similar results on tadpole growth and

development, and survival have been reported. This vast literature is not cited in this paper, but it should be. Perhaps those previous results are linked to reduced microbiota of tadpoles under stress, high density, or other conditions, and this could be an important future direction for such studies.

The authors note that:

Additionally, adult resistance to infection could not be explained by juvenile mass at metamorphosis (GLMM, $\chi^2 = 0.00$, $df = 1$, $P = 0.98$), days to metamorphosis ($\chi^2 = 0.19$, $df = 1$, $P = 0.66$), and adult mass ($\chi^2 = 0.92$, $df = 1$, $P = 0.34$).

They also show that adult resistance to worms established in the gut was affected by water treatment.

Thus,

water treatment affected growth and survival

water treatment affected microbiota

water treatment affected adult resistance.

Their structural equation modeling cannot rule out the confounding effect on growth, and indeed, they did not test these factors:

"SEMs were only conducted on tanks for which we had all the measured variables: juvenile bacterial diversity (gut and skin), adult gut bacterial diversity, and infection data."

No tadpole growth or survival data are included in their models.

Thus, I cannot agree with these results:

"Given the convergence of the microbiota in adults across water treatments, the effect of juvenile water treatments on adult host resistance must be mediated by the microbiota of juveniles. This conclusion is supported by our SEM, "

The authors have not shown that microbiota mediate the effects, only that water treatment mediated the effects.

I would say that microbiota is correlated with the effects.

Discussion:

I would change this:

"These results suggest that an early-life disruption of the host-associated microbiota affects the development of host resistance mechanisms, such as immunity, and has enduring effects on host resistance later in life."

to this:

These results suggest that an early-life disruption of the host-associated microbiota, or alternatively a disruption of tadpole growth or other side-effect of antibiotic exposure, affects the development of host resistance mechanisms, such as immunity, and has enduring effects on host resistance later in life.

Reviewers' comments (italics) and authors' responses (bold):

Reviewer #1 (Remarks to the Author):

The authors have responded to all comments made by the reviewers, and I have no further comments. I congratulate the authors for a job well done and hope to see this paper published in Nature Communications.

Reviewer #2 (Remarks to the Author):

The main conclusion is this: "Thus, we demonstrate that an early-life disruption of the microbiota had lasting reductions on host resistance to infections, which was likely mediated by its effects on immune system development." I remain unconvinced of this conclusion. Yes, the antibiotic treatment reduced bacteria in tadpoles. It also reduced growth rate and probably immune system development. So it would be just as accurate to say that reduced growth rate as larva, or antibiotic toxicity, had lasting reductions on host resistance to infections, and thus preventing early life disruption of growth, or preventing antibiotic toxicity, could affect long-term health. There are no treatments showing reduced microbiota independent of growth rate caused the same increase in infection as adults. There are no experiments showing how the gut microbiota of tadpoles are involved mechanistically in infection susceptibility as adults. Thus, I cannot accept the conclusion without a major caveat in the abstract, introduction, and discussion wherever this result is reported. A trade off between growth and (immune) development is widely known for developing amphibians. This study may be another demonstration of this, and have nothing to do with the microbiota.

Reviewer 2 is correct that the long-term antibiotic treatment reduced growth and survival of frogs. However, the other two manipulated water treatments, sterile pond water only and short-term antibiotics, affected the early-life microbiota and later-life resistance to infection without affecting growth or survival of frogs. Moreover, there were not even trends for these two manipulated water treatments to affect growth or survival (Supplemental Table 4). Thus, our main finding that an early-life disruption in host-associated microbiota affects later-life resistance to infection was not confounded by the effect of any treatment on growth or survival.

We re-wrote the sentences on lines 97 and 102 to clarify these results: "Juveniles from the long-term antibiotic treatment took twice as long to metamorphose and had lower survival compared to juveniles from the control treatment and other manipulated water treatments..." and "...adults exposed to long-term antibiotics weighed less than frogs from the control treatment and other manipulated water treatments..."

I appreciate the new first section of the results that gives details on the reduced growth rate of antibiotic treated tadpoles, reduced growth rate of juveniles that were previously antibiotic treated, and lower survival of antibiotic treated tadpoles. I believe these results are key to understanding and interpreting this study: "Effect of water treatment on host health. Water treatment significantly affected juvenile survival (Coxme, $\chi^2 = 16.69$, $df = 3$, $P < 0.001$; Fig. 1; Supplemental Table 2) and time to metamorphosis (GLMM, $\chi^2 = 120.85$, $df = 3$, $P < 0.0001$). Juveniles from the long-term antibiotic treatment took twice as long to metamorphose and had lower survival compared to juveniles from the other treatments (sequential Bonferroni post-hoc

multiple comparison test: $P < 0.05$ for long-term antibiotic frogs compared to all other treatments). ... juvenile water treatment significantly affected adult mass when measured two months after metamorphosis (GLMM, $\chi^2 = 19.44$, $df = 3$, $P < 0.001$); adults exposed to long-term antibiotics weighed less than frogs from the other treatments ($P < 0.05$ for long-term antibiotic frogs compared to all other treatments). " The lower survival is an indication that the animals are not in the best health, but this may or may not be because of a reduced microbiota as tadpoles. At high densities, or under other stressful conditions, similar results on tadpole growth and development, and survival have been reported. This vast literature is not cited in this paper, but it should be. Perhaps those previous results are linked to reduced microbiota of tadpoles under stress, high density, or other conditions, and this could be an important future direction for such studies.

As stated above, two out of the three manipulated water treatments, sterile pond water only and short-term antibiotics, affected the early-life microbiota and later-life resistance to infection without affecting survival and time to metamorphosis of frogs. Thus, tadpole density, due to survival or time to metamorphosis, was not a factor that could explain our results. We revised lines 97 and line 102 to clarify these results.

The authors note that: Additionally, adult resistance to infection could not be explained by juvenile mass at metamorphosis (GLMM, $\chi^2 = 0.00$, $df = 1$, $P = 0.98$), days to metamorphosis ($\chi^2 = 0.19$, $df = 1$, $P = 0.66$), and adult mass ($\chi^2 = 0.92$, $df = 1$, $P = 0.34$). They also show that adult resistance to worms established in the gut was affected by water treatment. Thus, water treatment affected growth and survival, water treatment affected microbiota, and water treatment affected adult resistance. Their structural equation modeling cannot rule out the confounding effect on growth, and indeed, they did not test these factors: "SEMs were only conducted on tanks for which we had all the measured variables: juvenile bacterial diversity (gut and skin), adult gut bacterial diversity, and infection data." No tadpole growth or survival data are included in their models. Thus, I cannot agree with these results: "Given the convergence of the microbiota in adults across water treatments, the effect of juvenile water treatments on adult host resistance must be mediated by the microbiota of juveniles. This conclusion is supported by our SEM, " The authors have not shown that microbiota mediate the effects, only that water treatment mediated the effects. I would say that microbiota is correlated with the effects.

We re-ran the structural equation model two additional ways. First, we excluded all samples from the long-term antibiotic treatment, which was the only treatment to show any evidence of effects on host growth or survival (Supplemental Table 2). Second, we included samples from the long-term antibiotic treatment but also additional models with the effect of adult mass and tank-level survival on adult resistance to parasites (Supplemental Table 3). We did not include an effect of metamorph mass on the microbiota because none of the water treatments had any discernible effects on metamorph mass (see Supplemental Table 4). In all of these additional analyses, the top model, as well as the significance of all pathways, remains the same as the original structural equation analyses.

Please see the tables below for the results of the additional analyses. We also included information regarding these analyses in the results of the main text on lines 137-143: "Because the long-term antibiotic treatment affected the growth and survival of frogs

(Supplemental Table 4), we included two additional SEM analyses: 1) without samples from the long-term antibiotic treatment (Supplemental Table 2), and 2) with samples from the long-term antibiotic treatment but also additional models with the effect of adult mass and tank-level survival on adult resistance to parasites (Supplemental Table 3). In these additional analyses, the top model remains the same as in the original SEM (Supplemental Table 1)."

Discussion:

I would change this:

"These results suggest that an early-life disruption of the host-associated microbiota affects the development of host resistance mechanisms, such as immunity, and has enduring effects on host resistance later in life."

to this:

These results suggest that an early-life disruption of the host-associated microbiota, or alternatively a disruption of tadpole growth or other side-effect of antibiotic exposure, affects the development of host resistance mechanisms, such as immunity, and has enduring effects on host resistance later in life.

Given our original analyses and additional analyses, we respectfully disagree with this change.

Supplemental Table 2. Ranked models (same methods and pathways as the original SEM) excluding samples from the long-term antibiotic treatment, which reduced growth and survival of frogs. Even without the long-term antibiotic treatment, the top-ranked models remain the same as the original SEM.

Model	Paths	AIC	ΔAIC	ω
1	a-b	289.57	0.00	0.88
2	c-f	293.57	3.99	0.12
3	a-b, i	350.03	60.45	0.00
6	a-b, i, k	357.36	67.78	0.00
4	a-b, h-i	357.39	67.81	0.00
7	a, k, i	364.95	75.37	0.00
5	a-b, h-i, k	367.71	78.31	0.00
8	c-f, h-j, l	381.08	91.51	0.00

Supplemental Table 3. Ranked models (same methods and pathways as the original SEM) including samples from the long-term antibiotic treatment but also two additional models with adult mass and overall tank survival, which were both lower in frogs from the long-term antibiotic treatment. Models 9 and 10 include the same pathways as models 1 and 2, respectively, from the original analysis but also the effect of adult mass and survival on adult resistance. The top ranked model is the same as the original SEM.

Model	Paths	AIC	ΔAIC	ω
1	a-b	333.15	0.00	0.72
9	a-b + mass and survival	335.79	2.64	0.19
2	c-f	337.97	4.82	0.07
10	c-f + mass and survival	340.75	7.60	0.02
3	a-b, i	409.97	76.72	0.00
6	a-b, i, k	415.13	81.98	0.00
4	a-b, h-i	415.21	82.06	0.00
5	a-b, h-i, k	422.59	89.44	0.00
7	a, k, i	425.21	92.06	0.00
8	c-f, h-j, l	436.91	103.76	0.00

REVIEWERS' COMMENTS:

Reviewer #2 (Remarks to the Author):

The authors have now clarified their results in this second revision. My previous comments were mostly based on the antibiotic treatment which affected growth and survival. In addition to clarifying the first section that was added to the results, "Effect of water treatment on host health," I would like to see a presentation of the data, not just the statistics for this section. A supplemental figure would suffice, showing the actual values for each treatment on the amphibian health. This would convince the reader that it really is the microbiota causing the later-life infection results, rather than any confound in growth or survival (at least for the sterile water no antibiotics, and short-term antibiotics treatments).

Reviewer's comment (italics) and authors' comment (bold):

Reviewer #2 (Remarks to the Author):

The authors have now clarified their results in this second revision. My previous comments were mostly based on the antibiotic treatment which affected growth and survival. In addition to clarifying the first section that was added to the results, "Effect of water treatment on host health," I would like to see a presentation of the data, not just the statistics for this section. A supplemental figure would suffice, showing the actual values for each treatment on the amphibian health. This would convince the reader that it really is the microbiota causing the later-life infection results, rather than any confound in growth or survival (at least for the sterile water no antibiotics, and short-term antibiotics treatments).

In the supplementary material, we present a summary table of these results (Supplementary Table 3). We also deposited the raw data on FigShare (doi: 10.6084/m9.figshare.5044825).